# Remote Monitoring of Chronic Critically Ill Patients after Hospital Discharge: A Systematic Review

**DOI:** 10.3390/jcm11041010

**Published:** 2022-02-15

**Authors:** Dmitriy Viderman, Elena Seri, Mina Aubakirova, Yerkin Abdildin, Rafael Badenes, Federico Bilotta

**Affiliations:** 1Department of Anesthesiology and Intensive Care, Nazarbayev University School of Medicine, Nur-Sultan 010000, Kazakhstan; dmitriy.viderman@nu.edu.kz; 2Department of Anesthesiology, Critical Care and Pain Medicine, Policlinico Umberto I, “Sapienza” University of Rome, 00185 Rome, Italy; eliseri2499@gmail.com (E.S.); bilotta@tiscali.it (F.B.); 3Department of Biomedical Sciences, Nazarbayev University School of Medicine, Nur-Sultan 010000, Kazakhstan; mina.aubakirova@nu.edu.kz; 4School of Engineering and Digital Sciences, Nazarbayev University, Nur-Sultan 010000, Kazakhstan; yerkin.abdildin@nu.edu.kz; 5Department of Anesthesiology and Surgical-Trauma Intensive Care, Hospital Clínic Universitari, University of Valencia, 46010 Valencia, Spain

**Keywords:** critical care, remote monitoring, remote neurological monitoring, glucose monitoring, chronic critical illness, long-term care

## Abstract

Background: Over the past few decades, critical care has seen many advancements. These advancements resulted in a considerable increase in the prevalence of chronically critically ill patients requiring prolonged medical care, which led to a massive increase in healthcare utilization. Methods: We performed a search for suitable articles using PubMed and Google Scholar from the inception of these databases to 15 May 2021. Results: Thirty-four articles were included in the review and analyzed. We described the following characteristics and problems with chronic critically ill patient management: the patient population, remote monitoring, the monitoring of physiological parameters in chronic critically ill patients, the anatomical location of sensors, the barriers to implementation, and the main technology-related issues. The main challenges in the management of these patients are (1) the shortage of caretakers, (2) the periodicity of vital function monitoring (e.g., episodic measuring of blood pressure leads to missing important critical events such as hypertension, hypotension, and hypoxia), and (3) failure to catch and manage critical physiological events at the right time, which can result in poor outcomes. Conclusions: The prevalence of critically ill patients is expected to grow. Technical solutions can greatly assist medical personnel and caregivers. Wearable devices can be used to monitor blood pressure, heart rate, pulse, respiratory rate, blood oxygen saturation, metabolism, and central nervous system function. The most important points that should be addressed in future studies are the performance of the remote monitoring systems, safety, clinical and economic outcomes, as well as the acceptance of the devices by patients, caretakers, and healthcare professionals.

## 1. Introduction

Over the past decades, critical care has made significant advancements that have improved the outcomes of critically ill patients. Unfortunately, these advancements contributed to a considerable increase in the prevalence of chronic critical illness requiring prolonged medical care, including mechanical ventilation, which has led to a massive increase in healthcare utilization [1].

Several definitions of chronic critical illness (CCI) have been proposed. One of the most accepted is the following: “a disease state which affects intensive care patients who have survived an initial insult but remain dependent on intensive care for a protracted period, neither dying nor recovering” [2].

Although the exact prevalence of CCI patients is difficult to estimate, it was calculated to be as high as 250,000 patients in the United States alone, and rapidly growing [3,4]. It is expected to grow 50–100% in every upcoming decade [3,4]. Healthcare spending related to chronic critical illness has been estimated to exceed $20 billion and is expected to rise further [1]. Furthermore, readmission rates within one year after hospital discharge have exceeded 40% [5]. Many patients who are discharged to long-term care facilities are unable to be adequately rehabilitated in order to return home within 6 months and are usually institutionalized until death [6]. An additional challenge for CCI patients is life quality and expectancy given that fewer than 12% of chronically critically ill patients were alive and independent one year after their acute illness [7,8,9,10]. The clinical course and management strategy of chronic critical illness are different from those of acute critical illness. Throughout the course of the chronic critical illness, the clinical status usually fluctuates slowly [11]. Improvement of condition and organ-system function takes place slowly and usually takes weeks or months to occur [11]. It is also important to ascertain that although chronic critical illness is a chronic process, it can include rapid acute events such as hemodynamic instability, heart failure, pneumonia, and sepsis that require an escalation of management [11]. The unpredictable clinical trajectory of chronic critical illness requires caregivers to have unique skills sets that should ideally combine rehabilitation, emergency, and sometimes critical care skills. Occasionally it is not possible to observe CCI patient’s conditions on a constant basis; therefore, technological support that monitors vital functions, detects functional deterioration, and possibly replaces or supports vital functions is required [11].

One of the central issues in the management of CCI patients after their discharge from the hospital is the absence of timely and continuous monitoring of even the most basic physiological parameters, and the lack of immediate correction and treatment when needed. After discharge from the hospital, these patients frequently have unstable vital functions [11]. Moreover, basic physiological parameters, such as blood pressure, pulse, respiratory rate, consciousness, and oxygen saturation, are measured episodically; therefore, important pathological events can be frequently missed. Failure to recognize these events can result in life-threatening complications and make all previous attempts by the medical team to manage the patient futile. Therefore, for more organized management, the creation of specialized facilities was proposed [11].

Remote patient monitoring suggests a promising direction in healthcare that could help in the diagnosis and treatment of patients remotely using sensor devices, telecommunication, and information technology solutions for the management of chronic critical illness. These solutions collect medical data directly from the patient and transmit the data to the caregivers and healthcare providers for interpretation and recommendations [12].

Remote patient monitoring may help to better control common conditions, their complications, and life-threatening events (e.g., metabolic, respiratory, cardiovascular, and neurological), therefore reducing the risk of hospitalization and mortality and improving the overall quality of life and services.

The purpose of this systematic review is to report evidence on the possible applications of wireless remote monitoring of chronic critical illness patients after hospital discharge with the aim of indicating a potential pathway for future research to develop evidence-based recommendations using cutting-edge remote monitoring technologies.

## 2. Materials and Methods

This systematic review was performed in accordance with the Preferred Reporting Items for Systematic Reviews and Meta-Analyses (PRISMA) statement [13]. The study was registered and accepted into the international prospective register of systematic review databases (PROSPERO registration number: CRD42021255515) [14].

We developed a protocol for matching publications which was established and approved by the research group. We predefined our research subtopics and performed a systematic review to summarize the current state of application of wireless remote monitoring of chronically critically ill patients after hospital discharge.

The steps in our evidence search and synthesis include: (1) Identification of relevant publications; (2) Data extraction; (3) Data analysis, aggregation, and summarization of the results; (4) Synthesis of the existing data; (5) Identifying the implications of the study findings; (6) Detailing the existing clinical gaps; (7) and Synthesis of conclusions.

We defined the scope of this review as all available articles reporting wearable technologies and sensors that might be used for chronically critically ill patients after discharge from the hospital.

Article selection: Articles were included in this systematic review if they mentioned the following: (1) 18 years and older; (2) Clearly described methodology of the study; (3) Applications of wireless remote monitoring of chronic critically ill patients after hospital discharge that can potentially be used for vital function monitoring (cardiovascular, respiratory, nervous system, temperature); (4) and Vital function monitoring.

Articles were excluded from the study if: (1) they did not clearly describe the study methodology; (2) they were animal studies; (3) and if they were pediatric studies (<18 years old).

Settings: Any healthcare setting (medical centers, hospitals, clinics).

Types of study to be included: all types of studies and reports should be included in accordance with inclusion. Both retrospective and prospective studies were considered.

Search methods: We performed a search for suitable articles using PubMed and Google Scholar from the inception of these databases to 15 May 2021. The searches included the following terms and their combinations: “chronic critical illness”, “critical illness”, “remote monitoring,” “outcome,” and “monitoring”. We searched the journals and references for all articles relevant to the study. Ethical approval and patient consent were not required. Since remote patient monitoring was not well studied in the chronically ill patient population after their discharge from the hospital, we identified the main syndromes and symptoms commonly seen in chronic critical illness (Figure 1) and searched for technologies that could potentially be used in this patient population. The following information was extracted: reference, first author, year of publication, study goals, study type, targeted population, age, gender, sample size, diagnosis, comorbidities, type of device, monitoring parameter (analysis), electroencephalogram (EEG), electrocardiogram (ECG), glucose level, outcomes, location of sensors, data processing, artificial intelligence method, performance of the models, and implementation barriers.

## 3. Results

### 3.1. Study Characteristics

A literature search yielded 1200 publications. After the removal of duplicates and the selection of studies that met the inclusion criteria, 34 articles were included in the review and analyzed (Figure 1) [9,15,16,17,18,19,20,21,22,23,24,25,26,27,28,29,30,31,32,33,34,35,36,37,38,39,40,41,42,43,44,45,46,47,48,49,50,51]. Data on the study characteristics, including author, year of publication, country, study objective and design, study population and sample size, patient age and gender distribution, type of device, location of sensor, parameter of monitoring, issues with devices and barriers to implementation, as well as study conclusions and findings are summarized in Table 1.

### 3.2. Patient Population

Chronically critically ill patients present a wide variety of symptoms and syndromes and may require medical and surgical treatments, including: (1) remote monitoring of infections, hemodynamics, sepsis, and pain; (2) remote monitoring of surgical care (neurosurgical, orthopedic, spinal, vascular, abdominal, transplant); (3) remote monitoring of patients undergoing acute or chronic neurologic care (including dementia); (4) remote monitoring of patients receiving opioids and other centrally acting central nervous system suppressants.

### 3.3. Remote Monitoring of Physiological Parameters in Chronic Critical Ill Patients

Since one of the most important issues in the management of such patients is the failure to continuously monitor physiological functions, the missing of critical events and late responses to derangements can lead to complications. The following physiological parameters have been monitored and reported in the published studies: non-invasive blood pressure, pulse, respiratory rate, SpO2, skin temperature, electrocardiogram, continuous noninvasive blood pressure, respiratory rate, pulse oximetry, temperature, body posture, fall detection, activity, step count, and ambulation (Table 1) [9,15,16,17,18,19,20,21,22,23,24,25,26,27,28,29,30,31,32,33,34,35,36,37,38,39,40,41,42,43,44,45,46,47].

### 3.4. Anatomical Location of the Sensor

The most common anatomical locations for sensors were the chest (sensors for respiratory rate, ECG), wrist, thumb (sensor for SpO2 and BP), thigh, calf, and lower arm.

Patient-related issues:

1. Devices were described as heavy, bulky, uncomfortable to wear, difficult to wear while performing activities, difficult to wear while eating, difficult to wear while washing hands; 2. Patients reported anxiety over possible injury and pressure sores; 3. Patients were concerned that remote monitoring will replace face-to-face interaction with nurses [7].

Technical issues of using remote monitoring devices:

1. Issues with responsiveness of the screen to input, robustness, and ease of cleaning the device; 2. Artifacts caused by connection failure [4], motion of the sensors, patient movements, and need for calibration of the physiological parameters such as blood pressure; 3. A large amount of data generated each day by a wearable device; 4. The digital patch was reported to be unsuitable for patients with atrial fibrillation [48]; 5. Monitoring devices deliver falsely reassuring data that may reduce the attention that patients require [48]; 6. The differences between vital sign patches and manual measurements of vital signs were out of acceptable limits [49]; 7. Signal quality can be affected by many factors such as inappropriate sensor-skin coupling due to device malposition, pressure on skin, ambient light, and biological factors and motions [50]; 8. The devices for remote monitoring require the development and improvement of interoperability standards to assist device connectivity and the integration of a monitor into medical settings [6]; 9. Data loss due to technical issues [49]; 10. Large volume of data related to cardio-respiratory function (heart rate/rhythm and respiratory rate cancelled by the system) [48,52]; 11. Redundant amount of data can produce “false-positive” outcomes and, therefore, should be double-checked carefully to warrant use of new technologies and approaches in trauma care [53].

### 3.5. Medical Professional-Related Concerns

1. Nurses were concerned that the devices would replace them [1,7,15]. 2. The high volume of data received from the monitoring devices can lead to an increased workload on personnel and could lead to personnel withholding themselves from checking these data, resulting in a diminution of the predictive value of continuous monitoring [4]. 3. A high quantity of redundant audible alerts interrupts nursing work and apparently reduces patient safety [11,54]. 4. Additional training for caregivers and healthcare professionals might be required (study coordinators requested to gain additional experience with devices and their software to increase their comfort in managing these devices) [55]. 5. Unnecessary false alarms may lead to the loss of attention of healthcare providers and caregivers to patients [2,12,16]. 6. The personnel did not always identify a deterioration pattern [3].

## 4. Discussion

This systematic review summarizes the evidence related to the application of remote monitoring of chronically critically ill patients after hospital discharge. Although there is an insufficient amount of evidence related to the home-based management of these patients using advanced technologies, we summarized the evidence from other areas of remote monitoring that can be used for this group of patients. Multisystem dysfunction, including cerebral, cognitive, and behavioral impairment is present in almost all patients with chronic critical illness. Most patients have functional impairment, which requires close observation and involvement of caregivers [5,7,8,10,51,56,57]. Therefore, the remote monitoring systems might be valuable for this patient population.

### 4.1. General Rationale of Using Remote Patient Monitoring

Remote monitoring systems include a monitor/terminal for the end-user, a communication network, a data acquisition system, and a data processing system. The incentives for remote monitoring include real-time and continuous tracking of symptoms, early detection of complications and deterioration of vital function, lower treatment costs, and the ability to activate an emergency response, if needed [55]. Implementation of remote patient monitoring can improve the quality of patient management, reduce complication and deterioration rates, and decrease the burden on family by likely decreasing healthcare costs through catching and responding to complications and deteriorations as early as possible.

Innovations in digital medicine are revolutionizing healthcare delivery and are significantly changing the interaction of healthcare providers with patients by developing and expanding the functions of monitoring devices [53,58]. The development of monitoring devices has led to opportunities for remote monitoring of clinically important physiological variables outside of hospital settings [59]. Such devices can be implemented into routine management of chronic medical conditions and can provide useful information for both medical personnel and patients [58,60]. A lot of attention has been given to the applications of wearable body sensors for remote monitoring [61].

A wide variety of sensors can be incorporated into smart devices to allow for the remote monitoring of the most appropriate variables and for data transmission. These sensors can measure numerous variables including vital signs (blood pressure, respiratory rate, temperature, level of consciousness, blood oxygen saturation, blood glucose level) and body movement. The sensors can be implanted in body parts and clothing, or subcutaneously. They are becoming more reliable, accurate, and easy to use for patient monitoring [62].

Such technologies can be used for continuous monitoring, prediction, prevention, diagnosis, and treatment of pathological events in chronic critical illness. Despite considerable advancement in this area, the widespread use of this technology in chronic critical illness remains very limited. The most important types of sensors that can be useful for such patients after hospital discharge can give measurements on hemoglobin oxygen saturation, heart rate, respiratory rate, ECG, blood glucose concentration, body temperature, posture and movement, vibrations, and coughing events. The wearable devices can be worn in several anatomical locations, such as the chest, arm, leg, waist, and wrist [58].

The most broadly accepted directions of remote monitoring include patient reported outcomes, telemonitoring, and quantifying self-hybrid models [50].

The applications of wearable sensors in the management of chronically critically ill patients might be useful away from the hospital. These sensors may reduce the length of hospital stay, cost of hospitalization, and hospital bed turnover. In turn, the use of wearable technologies might also improve the quality of patient care in non-hospital settings, reduce family burden, and likely prolong the life span and the quality of rehabilitation, reducing the risk of complications and readmission to hospitals.

Remote monitoring has been shown to increase the quality of care in cardiovascular patients [63]. Moreover, there is a high demand for device-driven detection of breathing patterns, respiratory rate, and fatal respiratory disorders [63]. The ring sensor devices have been used to improve the management of congestive heart failure and hypertension [58,64]. Vital sign monitoring is the most important type of monitoring that can measure a wide variety of parameters, ranging from electrical to biochemical signs [65].

### 4.2. Glucose Monitoring

Diabetes has been consistently reported as one of the most prevalent chronic conditions in chronic critically ill patients. To decrease the risk of further complications, strict glucose control is required. Blood glucose levels, especially in chronically ill patients with diabetes, can fluctuate significantly throughout the day and, therefore, may require multiple daily measurements. Traditional glucose control based on blood sample collection through a finger-prick is an invasive, inconvenient method [66]. The application of wearable sensors for measuring blood glucose levels could improve the quality of diabetes management and patients’ autonomy [47]. Wearable sensors with the function of continuous glucose monitoring have been successfully used to reduce hemoglobin A1c (HbA1c), improving the quality of life and health outcomes of patients [65]. The medium and long length electrodes penetrate into the deep layers of the tissue and give data related to fluctuations in glucose levels [58].

### 4.3. Remote Neurological Monitoring

Remote neurological monitoring plays an important role in the management of chronically critically ill patients, especially in outpatient postoperative management and rehabilitation. Several neurological parameters can be controlled using remote monitoring devices. A wristwatch has been shown to detect seven out of eight seizure episodes and accurately transfer related information to the caregiver [9]. While this device does not predict or treat convulsions, it can alert the caregiver quickly, reducing the risk of serious damage and death [58]. Although remote monitoring devices have been used for a wide variety of diseases and conditions, high-quality studies regarding chronic critical illness are missing. Given the progressively increasing number of such patients, it might be reasonable to implement these technologies in chronically critical ill patients after hospital discharge.

### 4.4. Limitations of This Study

One of the most important limitations of this systematic review is the high heterogeneity of the reported data. Since most of the included studies were performed by researchers with technical majors (engineering, computer sciences), most of them focused on technical characteristics, and not all studies reported the performance data that would help to generate clinical evidence (sensitivity, specificity, positive and negative predictive data). Therefore, we did not perform a meta-analysis. Despite this, we found it useful to transfer the existing knowledge from technical to medical fields. Another limitation is that since the definition of chronic critical illness is not widely used in clinical practice, the number of studies specifically focusing on chronic critical illness patients after hospital discharge is very limited. There was not enough available evidence to conduct the subgroup analysis, e.g., in some studies the sample size was too small, in some studies there were too many comparisons undertaken, and some other studies either did not characterize the patient population or did not subdivide them into groups at all. We did not break down the study sample of the secondary data into subgroups, due to the reason that we might end up with too few participants in each group to detect differences, or to ensure that differences were not a matter of chance. Therefore, we focused on the general issues and potential solutions in the management of chronic critically ill patients after the discharge from the hospital without subdividing the cohort into several groups.

Some of these limitations can be overcome by collaborating with medical doctors, clinical investigators, nurses, caregivers, engineers, and information technology professionals. This collaboration can minimize the existing gap and make clinical trials in this area available. Only successful clinical trials can lead towards a wide implementation of these technologies in clinical practice.

### 4.5. Future Research

It is expected that advanced wearable technologies will continue to evolve. We need to ensure that we will be able to provide appropriate care to CCI patients. There is not enough attention from both the healthcare and scientific worlds toward this problem. The responsibility of healthcare, science, and technology is to make the lives of patients easier and improve their quality. Chronic critical illness is a healthcare and societal issue that could be potentially solved by creating a special services and involving modern technologies.

The implementation of remote-sensor based monitoring technologies can bring the following benefits to the existing healthcare system: 1. Reduce the length of hospital stays; 2. Reduce healthcare spending; 3. Reduce the burden on families and healthcare providers during the post-hospital discharge period; 4. Improve the quality of life of patients as well as their family members; 5. Enhance research and innovation; 6. Improve the personalization of medical care; 7. Enhance the adoption of these monitoring technologies by healthcare providers and patients.

The feasibility of monitoring activity patterns of patients with very severe forms of their conditions warrants further research [64]. Apparently, there are considerable economic investments devoted to pharmacologic rather than technological innovation [9]. Continuous patient monitoring systems can be successfully accepted, implemented, and used only if they improve efficiency in identifying patient destabilization and if they do not increase the workload of healthcare providers. Therefore, more studies are needed to address this issue.

## 5. Conclusions

Given the current progress in intensive care medicine, the prevalence of chronic critical illness will continue to grow, leading to an increase in family burden, healthcare utilization, and economic costs. The main issues in the management of these patients are: the shortage of caretakers, the episodic nature of vital function monitoring (e.g., episodic measuring blood pressure leads to missing important critical events such as hypertension, hypotension, and hypoxia), and failure to catch and manage critical physiological events at the right time which can finally result in poor outcomes. Wearable devices can be used to monitor blood pressure, heart rate, pulse, respiratory rate, blood oxygen saturation, metabolism, and central nervous system function. While numerous studies have been conducted, there are still many questions to be answered. The most important of these revolve around the performance of the remote monitoring systems (sensitivity, specificity, positive predictive value, and negative predictive value), safety, clinical and economic outcomes, and acceptance of the devices by patients, caretakers, and healthcare professionals. Future clinical trials are warranted to investigate the value of remote monitoring in the management of critically ill patients.

## Figures and Tables

**Figure 1 jcm-11-01010-f001:**
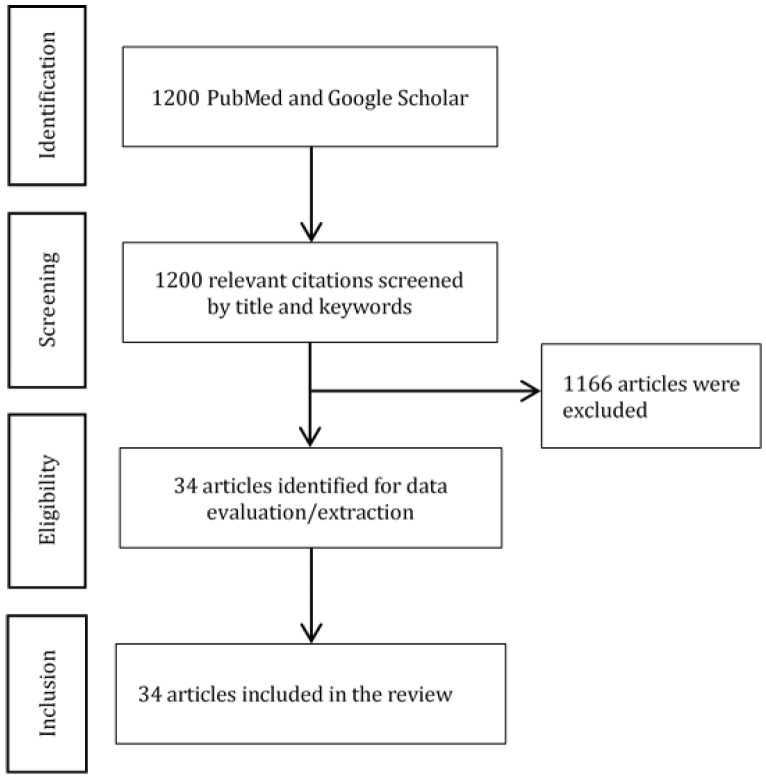
PRISMA diagram.

**Table 1 jcm-11-01010-t001:** Description of characteristics of included studies.

#	First Author, Year, Country, Reference	Objective (Study Type)	Targeted Population (*n*)	Gender (M/F, %)	Age	Device (Location of Sensor)	Parameter of Monitoring	Problems with Devices/Barriers to Implementation	Outcomes/Findings
1	Prgomet et al., 2016, Australia [15]	Clinical staff perceptions to monitoring practices (mixed methods)	Physicians, nurses (*n* = 106)	12.5/87.5	18–44 = 85% 45+ = 10%	ViSi Mobile (wrist)	BP, pulse, RR, SpO2, T, ECG	Inconvenience; technical issues; substitution of nurses with devices; false alarms	Positive expectations of CM on care improvement
2	Weller et al., 2017, USA [16]	Clinical outcomes under standard versus continuous VS monitoring with low alarm rates (case-control)	Older neuro- and neurosurgery patients (*n* = 736)	I: 54/46 C: 52/48	I: 60.5 (14.7) C: 60.1 (15.5)	ViSi Mobile (not reported)	BP, HR, RR, SpO2	Possibility of false alarms or overlooking real deterioration	CM was effective in detection of VS changes at a low alarm rate
3	Verrillo et al., 2018, USA [17]	Effects of using routine versus continuous VS surveillance (before-after)	Orthopedics, trauma (*n* = 864; I = 422, C = 427, Survey = 15)	I: 54/46, C: 58/42, Survey: 0/100	I: 54.45 (52.8–56.1) C: 51.44 (49.7–53.2) S: up to 29	ViSi Mobile (Chest, wrist, thumb)	HR, BP, RR, SpO2, T	None reported	CM allowed for improved detection of state exacerbation, lower complication rates, similar incidence of RRTs, reduced ICU requirement
4	Weenk et al., 2017, the Netherlands [18]	VS measurements by nurses versus two CM devices, and experience perceptions (mixed methods)	Internal medicine (sepsis, arthritis, BP control) and surgical patients (*n* = 20)	65/35	49.9 (13.4), range 33–82	ViSi Mobile (Chest, wrist, thumb), HealthPatch (Chest, wrist)	Visi Mobile: ECG, HR, SpO2, RR, T, and BP HealthPatch: ECG, HR, HRV, RR, T, body posture, fall detection, and activity	Artefacts due to technical issues, body motion, sensor detachment, and failure to carry the mobile device at all times. Skin irritation; inconvenience; detachment from skin; quick battery discharge; weak connectivity; large amount of data	Consistency in VS measured by both devices and manually. MEWS clinically significantly differed due to inconsistent RR measurements. Artifacts due to attachment issues and for undetected reasons. Positive attitudes.
5	Watkins et al., 2015, USA [19]	Evaluation of VS CM in hospital settings (prospective observational)	Patients and nurses in medical and surgical unit (*n* = 236 patients, *n* = 24 nurses)	NA	NA	ViSi Mobile (Not reported)	SpO_2_, HR, BP, RR	Possibility for excessive number of alarms	Feasibility of CM at a reasonable alarm rate
6	Downey et al., 2018, UK [20]	Evaluation of VS CM practicality for surgery patients (pilot RCT)	Surgical patients (*n* = 350)	54/46	65.2, 24–94	SensiumVitals (Chest)	HR, RR, T	Excessive number of alerts before parameter resets. Various levels of involvement among nurses	Faster reception of antibiotics for sepsis, less time of hospitalization, lower 30-day readmission rates, higher perception of feeling comfortable and safe for the CM group
7	Downey et al., 2018, UK [21]	Patients’ perceptions of in-hospital CM (Qualitative)	Surgical patients (*n* = 12)	50/50	42–83	SensiumVitals (Chest)	HR, RR, T	Unpractical and not comfortable. Worry that the devices are not reliable and will substitute medical staff	CM perceived as valuable, especially at night, but lacking personal communication and unable to clarify health-related uncertainties
8	Hernandez-Silveira et al., 2015, UK [22]	Comparison of measurements between CM device and bedside monitor (Validation)	Elective surgery (1) and general ward (2) patients (*n* = 61; 1 = 20; 2 = 41)	1: 65/35 2: 78/22	1 = 49 (16)	SensiumVitals (Chest)	HR, RR, T	Not reliable for patients with atrial fibrillation. False negatives may result in lack of attention	Acceptable consistency of measurements between the CM device and bedside monitor: 80% for HR and 50% for RR
9	Hernandez-Silveira et al., 2015, UK [23]	Demonstration of practicality of a CM device in a hospital (Validation)	Patient simulators (1); healthy volunteers (2); clinical patients acute (3) (1 = 333; 2: first stage = 21, second stage = 6; 3 = 41)	1, 3: NA; 2: first stage = 86/14, second stage = 83/17	1 = NA; 2: first stage = 32.1 (6.9), second stage = 34.1 (11.6); 3 = 18–85	SensiumVital (Chest)	HR, RR, T	High rate of rejections in RR data for clinical patients	Satisfactory agreement between of measurements with a clinically approved bedside monitor.
10	Downey et al., 2019, UK [24]	Validation of accuracy of HR, RR, and T measurements by a CM device (Validation)	Post-operative patients (*n* = 51)	Not reported	Not reported	SensiumVitals (Chest)	HR, RR, T	RR artefacts possibly due to speaking. Differences in VS measurements by CM device and manually	Moderate correlations between measurements for HR (with large discrepancies), low correlations for RR and T
11	Chan et al., 2013, USA [25]	Analysis of performance of a CM device (Validation)	Older (1) and younger (2) healthy adults (*n* = 35; 1 = 15; 2 = 10)	1: 47/53 2: 50/50	1: 70 (5), 63–79 2: 25 (3.6), 18–29	Bluetooth Low Energy (BLE) (Over ICS 2 or 6 or over the upper sternum)	HR, HRV, RR, posture, steps, falls	Need for user-friendliness for wider acceptability	CM devices produce similar observations as standard and more bulky equipment
12	Izmailova et al., 2019, USA [26]	Evaluation of measurements of VS and physical activity by two CM devices (Validation)	Healthy adults (*n* = 6)	83/17	18–55	Actiwatch Spectrum Pro (A) (wrist); Vitalconnect HealthPatch MD (HP) (left upper precordium)	A: mobility and sleep HP: HR, RR, T	Poor correlation with hospital measurements, false signal of tachycardia, time-consuming to double-check	HealthPatch showed a strong correlation for HR, but not for RR or T, with manual measurements. Actiwatch found acceptable for physical activity/sleep surveillance and for assistance in interpreting VS data
13	Breteler et al., 2018, The Netherlands [27]	Realiability of HR and RR measurements by a CM device (Observational comparisons)	Post-surgery patients (*n* = 25)	72/28	63 (57.8–71.5)	HealthPatch MD (Chest)	ECG, HR, HRV, RR, T, posture, steps	Missing data due to unstable battery. Possible need to manually delete artefacts	Accurate measurements for HR but not for RR
14	Selvaraj et al., 2018, USA [28]	Presentation and lab validation of a CM device (Validation)	Healthy volunteers (*n* = 57)	58/42	35 (11)	VitalPatch (Chest)	HR, BR, posture, steps, and falls	None reported	Accurate measurements of VS and rest-activity cycles
15	Liu et al., 2014, USA [29]	Usefulness of a wireless CM device in ER for LSI (Validation)	Code 2/3 trauma (*n* = 305; C = 201; I = 104)	Overall 66/34	Overall 39 (16)	The wireless vital signs monitor (WVSM) (arm, thumb)	ECG, BP, SpO2	Human error during attachment to the patient. Training of medical staff, adaptation of medical settings to the device	Improvement in LSI using CM device in ER settings
16	Liu et al., 2015, USA [30]	Assessment of VS data quality of a wireless CM device and its ability to forecast requirement of LSIs (Cohort)	Code 2/3 trauma (*n* = 104)	79/21	40 (16)	WVSM (arm, thumb)	HR, BP, MAP, RR, SpO2, shock index, pulse pressure	Possibility for false-positive observations	Useful for forecasting LSI requirement, the majority of data being high quality
17	Razjouan et al., 2017, USA [31]	Effectiveness of a CM device to predict risk of fall (Cohort)	Hematology and oncology (*n* = 31)	45/55	59.5 (16.1)	Zephyr BioPatch (Chest)	ECG, RR, T, 3-dimensional acceleration	None reported	Risk of fall can be predicted by monitoring sleep and activity patterns and HRV
18	Boatin et al., 2016, USA [32]	Usefulness and patient experiences of a VS device (Mixed methods)	Pregnant women (1) (*n* = 32), Nurses (2) (*n* = 6)	0/100	1: 33.1 (9.7), 2: 33.5 (11)	BioPatch (Chest)	HR, RR, T	Minor discomfort	Useful for VS surveillance in pregnant women. Positive attitudes of patients and nurses
19	Kim et al., 2012, USA [33]	Comparison of CM measurements during physical activity in extreme temperatures with spirometry and mobile metabolic system (Validation)	Healthy individuals (*n* = 12)	100/0	25.5 (4.1)	BioHarness (Chest)	HR, RR	Artefacts due to motion and perspiration	Similar measurements during exercise between CM device and standard methods. Correlation high for HR, lower for RR
20	Van Haren et al., 2013, USA [34]	Assessment of the ability of MF to forecast LSI in prehospital settings (Cohort)	Trauma (*n* = 96, No LSI (1) *n* = 48, LSI (2) *n* = 48)	1: 88/12 2: 77/23	Overall 48 (19) 1: 47 (18) 2: 49 (20)	MWVSM (Forehead or limb)	T, SpO2, HR, pulse wave transit time	Occasionally poor connection	Useful in prehospital care for trauma patients
21	Meisozo et al., 2016, USA [35]	Comparison of a CM device in VS surveillance with standard hospital equipment (Cohort)	Trauma ICU patients (*n* = 59)	80/20	47 (20)	MWVSM (Forehead or limb)	BP, T, HR, SpO2	Data loss; under/over-triaging due to signal inaccuracy; requires improvements	In its current state, unreliable in identifying patients of highest medical priority
22	Dur et al., 2019, USA [36]	Accuracy of measurements and quality of signal (Observational)	Healthy (*n* = 35)	54/46	25 (4)	Wavelet Wristband (Wrist)	HR, HRV, RR	Quality of signal influenced by external aspects (movements, temperature, light, etc.)	Accurate measurements at rest
23	Li et al., 2019, USA [37]	CM device with capnography (Prospective pilot)	Respiratory patients in ER (*n* = 17)	59/41	Mean = 61	Philips wearable biosensor (Chest)	RR, HR, ambulation, posture	None reported	CM device is comparable in RR measurements with capnography in ER settings
24	Ordonnel et al., 2019, UK [38]	Extraction of sleep-wake activity data in patients of various degrees of disease severity (Cohort)	Heart failure (HF) patients (*n* = 11)	36/64	79 (8.3)	Proteus patch (Chest)	T, skin impedance, HR, RR	Unclear sleep-wake information in severe-condition patients	Feasible to monitor activity during sleep and wake time in HF patients
25	Hubner et al., 2015, Austria [39]	Effectiveness to identify priority cases (Observational cross-sectional)	ER patients (*n* = 226)	55/45	55 (43–71)	Philips IntelliVue Guardian Solution (Chest, arm, finger)	SpO2, pulse, RR, BP	Discomfort	Assists in identifying priority patients in ER. Positive attitudes.
26	Liu et al., 2013, China [40]	Evaluation of VS CM at rest and during exercise (Validation)	Healthy (*n* = 6)	100/0	22.3 (3.2)	EQ02 LifeMonitor (many possible locations)	HR, HRV, RR, ECG, RIP, body position, 3-axial acceleration	Costly due to non-reusability	Measurements are valid and reliable
27	Paul et al., 2019, Canada [41]	Clinical effectiveness and patient and staff experiences (Pilot RCT)	Surgery patients (I = 124, C = 126)	I: 24/76 C: 39/61	I: 58.0 (13.9) C: 57.5 (15.8)	Covidien Alarm Management System (finger)	SpO2, HR	False alarms due to technical issues; excessive alarms in tachycardic patients	Acceptable recruitment rate and positive experience
28	Pedone, 2013, Italy [42]	Effectiveness of telemonitoring COPD patients to decrease hospitalizations (RCT)	Elderly COPD stage II/III (*n* = 99, I = 50, C = 49)	I: 72/28 C: 63/37	I = 74.1 (6.4) C = 75.4 (6.7)	SweetAge (wrist)	HR, physical activity, T, galvanic skin response	None reported	Timely detection of state deterioration to allow for planned hospitalization
29	Pedone, 2015, Italy [43]	Effectiveness of tele-surveillance of VS (RCT)	Elderly with HF (*n* = 90, I = 47, C = 43)	I: 47/53 C = 30/70	I = 79.9 (6.8) C = 79.7 (7.8)	Sphygmomanometer, a scale, a pulse oximeter	SpO2, HR, BP	None reported	Tele-surveillance of VS decreases risk of hospitalization and all-cause mortality in elderly with HF
30	Chau, 2012, China [44]	Practicability and attitudes toward medical teleservices (RCT)	Elderly with COPD and hospitalization in the past year (*n* = 40)	97/3	72.93 (6.04)	Device kit (chest, finger)	SpO2, pulse, RR	Challenging for the elderly to read small screens, use multiple devices, often recharge battery	Positive patient perceptions
31	Dellaca, 2011 Spain [45]	Practicability of continuous positive airway pressure (CPAP) titration at home (Observational)	SAHS patients (*n* = 20)	56 (3)	NA	Autoset Spirit CPAP machine (mask)	Nasal pressure, breathing flow and air leak signals	Connection issues	Possibility for successful remote CPAP titration on patients with sleep apnea in home environment
32	Fox, 2012 Canada [46]	Improvement in adhering to PAP with telemedical surveillance (RCT)	Obstructive sleep apnea patients (*n* = 75, I = 39, C = 36)	I: 82/28 C: 78/22	53.5 (11.2) I: 52.0 (1.8) C: 55.2 (11.5)	EncoreAnywhere (mask)	PAP adherence, applied PAP pressure, mask leak, and residual respiratory events	Occasional side effects	Improved adherence to PAP with telemedical surveillance introduced at an early stage of treatment
33	Leelarathna et al., 2013, UK [47]	Evaluation of glucose CM device with two calibration methods in critically ill patients (RCT)	Patients with elevated insulin level (*n* = 24, I = 12, C = 12)	I: 75/25 C: 75/25	I: 62.8 (16) C: 58.3 (12.5)	FreeStyle Navigator (Subcutaneous)	Arterial blood glucose	None reported	Accurate CM of glucose, may be useful for intensive insulin therapy
34	Lockman et al., 2011, USA [9]	Identifying tonic-clonic seizures with a CM device (Cohort)	Epilepsy patients (*n* = 40, seizures = 6)	Seizures: 50/50	31 (23–48)	SmartWatch (wrist or ankle)	Rhythmic, repetitive movement of an extremity	Battery; connection	Measurements comparable to those of standard equipment

Abbreviation: BP—Blood Pressure; BR—Breathing Rate, C—Control, CM—Continuous Monitoring; COPD—Chronic obstructive pulmonary disease, CPAP—Continuous Positive Airway Pressure; ECG—Electrocardiogram; ER—Emergency Room; HF—Heart Failure; HR—Heart Rate, I—Intervention; ICU—Intensive Care Unit; RR—Respiratory Rate, T—Temperature, LSI—Life Saving Intervention; MAP—Mean Arterial Pressure; MEWS—Modified Early Warning Score; MF—Murphy Factor, PAP—Positive Airway Pressure; RCT—Randomized Controlled Trial; RIP—Respiratory Inductance Plethysmography, SAHS—Sleep Apnea-Hypopnea Syndrome; SpO2—Oxygen Saturation; VS—Vital Signs.

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
