# Peer review of "Remote Monitoring of Chronic Critically Ill Patients after Hospital Discharge: A Systematic Review"

_jcm, 2022, doi:10.3390/jcm11041010_

Round 1
Reviewer 1 Report
In the manuscript submitted by Viderman et al., the authors review the advantages and limitations of remote monitoring of chronic critically ill patients. The data from 34 pertinent manuscripts are presented in tabular form to include devices utilized, problems, and outcomes. Altogether, this is a timely and interesting review of burgeoning field of remote monitoring.
Comments:
1) It is suggested that the authors speculate on what might be the most advantageous instrument to be used for remote monitoring of the CCI patient. What specific parameters would be of the most importance to the monitoring care-giver.
2) Table 1 provides an extensive list of issues with the different instruments. The authors should provide some discussion on these issues and whether methods to overcome this difficulties could be invented.
Author Response
In the manuscript submitted by Viderman et al., the authors review the advantages and limitations of remote monitoring of chronic critically ill patients. The data from 34 pertinent manuscripts are presented in tabular form to include devices utilized, problems, and outcomes. Altogether, this is a timely and interesting review of burgeoning field of remote monitoring.
Comments:
1) It is suggested that the authors speculate on what might be the most advantageous instrument to be used for remote monitoring of the CCI patient. What specific parameters would be of the most importance to the monitoring care-giver.
2) Table 1 provides an extensive list of issues with the different instruments. The authors should provide some discussion on these issues and whether methods to overcome this difficulties could be invented.
Reply:
We would like to thank the reviewer 1 very much for the very useful comments.
We added the possible solutions to the reported limitations.
“These limitations can be overcome by collaborating medical doctors, clinical investigators, nurses, caregivers, engineers, information technology professionals. This collaboration can minimize the existing gap and make clinical trials in this area available. Finally, only successful clinical trials can lead towards a wide implementation of these technologies in the clinical practice”.
“Continuous patient monitoring system can be successfully accepted, implemented, and used only if it improves efficiency in identifying patient destabilization and if it does excessive workload to healthcare providers; therefore, more studies are needed to address this issue”.
Reviewer 2 Report
While reading the review, some issues appeared in the design, definition, reporting, and interpretation of findings, some of which are of major concern.
- The title, abstract and interpretation does not represent or specifically consider the studied population (brain injured chronic critical illness patients). Furthermore, brain injury is not specified by certain pathologies. A subgroup analysis would be advisable in order to provide a more specific scope and improve the validity of findings, the lack of which is mentioned as one of the limitations.
- The methods section mentions use of the PRISMA guidelines and PROSPERO, which are not referenced. Furthermore, the detail of how search terms were combined is described in insufficient detail and inclusion/exclusion criteria are incomplete (e.g. exclusion of articles in other languages). Articles of different patient populations were included in the review, which does not match the purpose of the review, focussing on patients with brain damage. It remains unclear what is meant by a manual search of databases, this hints towards an unsystematic fashion of review.
- Some of the statements made and numbers mentioned, especially in the introduction and discussion sections, are not contextualized well, may generalize evidence inadequately, or are scientifically inconclusive. For example, the statement in lines 40-44 (page 1), 72-73 (page 2) and 80-83 (page 2). They are also missing appropriate referencing. Discussion section should be condensed.
- Figures need major revision as arrows are not aligned, numbers are incorrect, some of the screenshots appear to be directly taken from MS Word and show words underlined in red. Labelling of tables is incorrect. A list of abbreviations would be advisable.
- The manuscript should be proofread by an English native speaker, some of the words and phrasing are inappropriate (“critical care science”). Punctuation needs to be consistent.
Author Response
While reading the review, some issues appeared in the design, definition, reporting, and interpretation of findings, some of which are of major concern.
- The title, abstract and interpretation does not represent or specifically consider the studied population (brain injured chronic critical illness patients). Furthermore, brain injury is not specified by certain pathologies. A subgroup analysis would be advisable in order to provide a more specific scope and improve the validity of findings, the lack of which is mentioned as one of the limitations.
Reply:
Dear Reviewer,
Thank you very much for your comments that substantially helped us to improve the quality of our manuscript. Unfortunately, it is impossible to conduct the subgroup analysis to date due to insufficiency of evidence. In the current manuscript, we discussed the general issues and potential solutions in the management of chronic critically ill patients after the discharge from the hospital. We tried to highlight the general approach of such patients with potential technological support.
- The methods section mentions use of the PRISMA guidelines and PROSPERO, which are not referenced. Furthermore, the detail of how search terms were combined is described in insufficient detail and inclusion/exclusion criteria are incomplete (e.g. exclusion of articles in other languages). Articles of different patient populations were included in the review, which does not match the purpose of the review, focusing on patients with brain damage. It remains unclear what is meant by a manual search of databases, this hints towards an unsystematic fashion of review. Reply:
We referenced the PRISMA guidelines, updated the search strategy, inclusion and exclusion criteria.
We also removed the word “manual” from the methods. We decided to focus on the general group of chronic critically ill patients after the discharge from the hospital, not just neurocritical patients.
- Some of the statements made and numbers mentioned, especially in the introduction and discussion sections, are not contextualized well, may generalize evidence inadequately, or are scientifically inconclusive. For example, the statement in lines 40-44 (page 1), 72-73 (page 2) and 80-83 (page 2). They are also missing appropriate referencing. Discussion section should be condensed.
Reply: We added the references to all the statements. - Figures need major revision as arrows are not aligned, numbers are incorrect, some of the screenshots appear to be directly taken from MS Word and show words underlined in red. Labelling of tables is incorrect. A list of abbreviations would be advisable.
-Reply: we revised the figures and added abbreviations.
We added the following abbreviations.
BP – Blood Pressure; RR – Respiratory Rate, C – Control, CM – Continuous Monitoring; COPD - Chronic obstructive pulmonary disease, CPAP - Continuous Positive Airway Pressure; ECG – Electrocardiogram; ER – Emergency Room; HF – Heart Failure; HR – Heart Rate, I – Intervention; ICU – Intensive Care Unit; RR – Respiratory Rate, T - Temperature, LSI – Life Saving Intervention; MAP – mean arterial pressure; MEWS - Modified Early Warning Score; MF – Murphy Factor, PAP – Positive Airway Pressure; RCT – Randomized Controlled Trial; RIP - respiratory inductance plethysmography, SAHS - Sleep Apnea-Hypopnea Syndrome; SpO2 - Oxygen saturation; VS – Vital Signs
- The manuscript should be proofread by an English native speaker, some of the words and phrasing are inappropriate (“critical care science”). Punctuation needs to be consistent.
Reply: We checked the grammar with assistance of a native English speaker. The phrasing is revised as well.